# Reconstruction of the transcriptional regulatory networks in the kidney of desert-adapted species
Fernando Alvira-Iraizoz [1,2] ✉, Benjamin T. Gillard [1], Audrys G. Pauža [1], Panjiao Lin[1,3], Alex Paterson[1], Pamela A. Burger [4], Mahmoud Hag Ali[5], Nabil Amor [6], Abdulaziz N. Alagaili[6], Abdu Adem [5,7] ✉, David Murphy [1,7] ✉ & Michael P. Greenwood [1,7]

Desert animals have evolved systems that enable them to thrive under dry conditions. A plethora of adaptations are well studied and, more recently, the molecular underlying mechanisms have been investigated, but these are not fully understood. We recently characterised the kidney transcriptomic adaptations of camels and jerboas that enable these animals to withstand water deprivation, revealing a role for cholesterol in camels. Here, we aim to further mine these data and run reconstruction of transcriptional regulatory networks (TRN) analyses in camel, jerboa and olive mouse kidneys under different water regimes. We identify a total of 283 differentially expressed transcription factors (TFs) and 209 regulatory networks among all species. Among them, we find *SREBF1* and *SREBF2*, which are key TFs involved in the cholesterol biosynthesis pathway. We further explore this pathway using immunofluorescence imaging. Localisation and/or signal of SREBP2 seems to change across conditions in camels and jerboas, perhaps indicative of protein retention in the Golgi during dehydration. Together, these data confirm a role for cholesterol during dehydration. Moreover, reconstruction of TRN analyses reveal a new set of potentially interesting genes and networks that would otherwise be ignored, these include but are not limited to other genes involved in cholesterol metabolism.

Climate change-associated desertification and extreme drought are posing an unprecedented burden on desert-dwelling species. However, some species have evolved to thrive under these conditions, and a plethora of adaptations to life in the desert have been described in them. While the physiology of these adaptations has been investigated for decades, the molecular and genomic underlying mechanisms are still a matter of ongoing investigations, since they are not fully understood yet. So far, multiple mechanisms and metabolic pathways have been associated with better adaptation to water scarcity and extreme environmental conditions, but others remain unclear.

We recently characterised the transcriptomes of the kidney of the one-humped Arabian camel (Camelus dromedarius) and the lesser Egyptian jerboa (Jaculus jaculus) during chronic dehydration and subsequent acute/

opportunistic rehydration[1,2]. Analyses of the gene expression data identified multiple genes involved in the water conservation machinery of the kidney that were differentially expressed (DE) under these conditions compared to controls. Further analyses of the transcriptome, proteome, as well as quantification and imaging of the cell membrane cholesterol revealed a role for cholesterol during chronic dehydration not previously described. Most of these changes returned to control levels with rehydration, whilst a different subset of DE genes were changed by rehydration compared to controls. Moreover, earlier studies on the olive mouse (Abrothrix olivacea)[3] showed differences in the transcriptomes of the kidney between a wild sub-population of animals adapted to xeric environments (the Patagonian steppe) and other mammalian models. Considerable differences in gene expression were also shown between this sub-population and another one

[1]Molecular Neuroendocrinology Research Group, Bristol Medical School: Translational Health Sciences, University of Bristol, Dorothy Hodgkin Building, Bristol, United Kingdom. [2]Genomics Medicine Unit, Navarrabiomed – Biomedical Research Centre, Pamplona, Spain. [3]Double-crane Biotechnology Co., Ltd. Medical Science and Technology Centre, Zhongguancun Life and Science Park, Changping District, Beijing, China. [4]Department of Interdisciplinary Life Sciences, Research Institute of Wildlife Ecology, Vetmeduni Vienna, Vienna, Austria. [5]Department of Pharmacology and Therapeutics, College of Medicine & Health Sciences, Khalifa University, Abu Dhabi, United Arab Emirates. [6]Department of Zoology, King Saud University, Riyadh, Kingdom of Saudi Arabia. [7]These authors jointly supervised this work: Abdu Adem, David Murphy, Michael P. Greenwood. ✉e-mail: f.alvirairaizoz@gmail.com; abdu.adem@ku.ac.ae; d.murphy@bristol.ac.uk

inhabiting the Valdivian and Magellanic rainforests. Lastly, similar studies have been carried out in other desert-adapted species, such as the cactus mouse (*Peromyscus eremicus*), which further showed how the kidney transcriptome is modulated to cope with water deprivation in multiple ways[4]. Despite the substantial contribution of these studies to our understanding of the field, a unified analysis is currently unavailable.

Transcriptional Regulatory Network (TRN) methodologies analyse gene expression data (e.g., transcriptomes) with reference to physical and regulatory interactions between genes (targets) and their regulators (Transcription Factors, TFs)[5,6]. TFs bind to cis-acting DNA elements associated with genes through a DNA-binding domain and either enhance or repress gene expression of their targets, but their own activity is usually regulated prior to binding the genomic DNA fragment by a variety of post-translational modifications, such as ligand-TF binding or phosphorylation. Moreover, TFs are the link between second messenger signalling pathways and gene expression[7]. Therefore, they are key regulators of cellular processes and typically regulate multiple target genes (TRNs or regulons). Thus, TRNs are paramount in development, differentiation and responding to environmental cues, and their misregulation may result in disease[8]. In fact, TRNs have become a key tool for the study of complex biological processes[9], and understanding TF dynamics in depth is essential to understand responses to environmental cues[7].

We aimed to explore the TRNs involved in chronic dehydration in desert-adapted species. Thus, we present here a reconstruction of the TRNs in the kidneys of the camel, jerboa, and olive mouse, which have been subjected to alterations in water availability. We revealed several TFs and significantly enriched TRNs that may be important to cope with desert environmental conditions. Moreover, based on our previous work, we validated the main TFs involved in regulating the synthesis of cholesterol, further investigated their transcript and protein dynamics, and reconstructed their regulatory networks. Our data shows that several TRNs are significantly enriched during dehydration, despite the TFs and targets forming these networks not necessarily reaching the differential expression significance threshold in the RNAseq analysis, thus opening a complete new set of potentially interesting genes that would otherwise be ignored. Further, we demonstrate that the regulatory machinery of the cholesterol synthesis pathway is affected at transcript and protein levels in the kidney of the one-humped Arabian camel and the jerboa during chronic dehydration, which supports our previous findings.

## Results

### Reconstruction of the TRN of Arabian camel, jerboa, and olive mouse

Using biomaRt, we extracted all the up-to-date assigned GO categories for every DE gene in our RNAseq data. Using GO:0003700 (DNA-binding transcription factor activity) to filter genes of interest, we identified a total of 49 TFs that were DE ($p_{adj} < 0.05$) in the kidney of the one-humped Arabian camel. Specifically, 15 TFs were DE in the medulla, 27 in the cortex and 7 in both tissues (Fig. 1a and Supplementary Data SD1). In the medulla, 7 TFs were DE in dehydration compared to controls, 4 in rehydration compared to controls and 18 in rehydration compared to dehydration, with a subset being co-regulated in more than one condition (Fig. 1b and Supplementary Data SD1). Similarly, in the cortex, 19 TFs were DE in dehydration compared to controls, 4 in rehydration compared to controls and 21 in rehydration compared to dehydration. Again, 10 TFs were co-regulated in more than one condition (Fig. 1c and Supplementary Data SD1). We further investigated the TRN of the camel kidney running pairwise comparisons of the three treatment groups using the Reconstruction of the TRN (RTN) package. Thus, we identified 36 regulons with significant ($p_{adj} < 0.05$) differential enrichment scores (dES) in the kidney of the one-humped Arabian camel (Supplementary Fig. SF1 and Supplementary Data SD2). We characterised regulons that were both significantly repressed (negative enrichment score), with their target genes overrepresented amongst those more negatively DE, and activated (positive enrichment score), with their target genes overrepresented among the genes with more positive log2 fold

changes. Interestingly, none of the TFs that control these regulons, nor any of their targets, showed significant changes in the RNAseq data.

Using the same method, we identified 211 DE TFs in the kidney of the jerboa. In this case, 74 TFs were DE in dehydration compared to controls, 186 in rehydration compared to controls and 124 in rehydration compared to dehydration. Amongst these, 152 TFs were co-regulated in more than one condition (Fig. 1d and Supplementary Data SD1). Moreover, RTN analyses revealed 138 significantly enriched ($p_{adj} < 0.05$) regulons after pair-wise comparisons of the three conditions (Supplementary Fig. SF2, SF3 and SF4; and Supplementary Data SD2). We identified a DE ($p_{adj} < 0.05$) regulon for 105 out of 211 DE TFs. Contrary, 32 regulons were significantly ($p_{adj} < 0.05$) enriched despite their master regulator not being DE. In this case, most of the regulons were suppressed. Interestingly, GO analysis of genes in the enriched TRNs (regulons), the TFs of which were also identified as DE, revealed pathways linked to response to stress, homoeostasis processes, inflammatory response and glucose homoeostasis.

In the olive mouse, we identified 74 DE TFs in the kidney of animals living in xeric conditions (Patagonian steppe) compared to those inhabiting humid areas (Valdivian and Magellanic rain forests) (Fig. 1e and Supplementary Data SD1). Further, we were able to characterise 68 significantly enriched ($p_{adj} < 0.05$) regulons. Of these, 27 regulons were activated whilst 41 were suppressed (Supplementary Fig. SF5 and Supplementary Data SD2). RTN revealed that all TFs regulating these regulons, except 3, were also DE. Thus, the RTN algorithm was able to characterise three additional significantly enriched regulons. GO analyses showed that these regulons did not seem to be linked to canonical pathways associated with water conservation.

Interestingly, we detected regulons being partially co-regulated by 2 or more TFs as shown by the fact that different regulons are composed by very similar gene sets. Further, the same regulon may be enriched under 2 different treatments, but the gene subset regulated under each condition might differ. This may indicate directionality in the response to opposite stimuli. Although most of the TFs associated with significantly enriched regulons were DE according to our RNAseq data, using RTN analyses enabled us to detect regulons which TFs were not DE and, thus, they could only be detected using this method. Further, these analyses revealed information about gene regulation, adding value to a gene-by-gene approach.

A cross-species comparison using mouse orthologs shows that 17, 24 and 7 DE TFs are shared between camels and jerboas, jerboas and olive mice and camels and olive mice, respectively. Only 1 DE TFs (Zbtb17) is shared amongst the 3 species (Fig. 1f and Supplementary Data SD1). Regarding significantly enriched regulons, the majority were species-specific, whilst only 24 were described in more than one species (Fig. 1g and Supplementary Data SD2). It is worth noting that the numbers in Fig. 1f, g do not match exactly those of preceding panels. The reason for this is that using mouse orthologues for the comparison inevitably introduces duplicates and results in unannotated genes being discarded due to the incompleteness of the camel and jerboa reference genomes annotations. We also ran this comparison using gene names, but in that case, many duplications are generated for the same reason, thus introducing a similar bias. In order to be more stringent, we decided to use Mus musculus Ensemble IDs that are less sensitive to poor annotation. It is important to note that this is not an attempt to pair-wise compare the three species, since they are fairly different in many ways, but a way to depict TFs and pathways potentially conserved in species adapted to water deprivation.

### Transcriptional regulation of the cholesterol biosynthesis process during water deprivation in camels and jerboa

Based on our previous research[2], where we demonstrated that cholesterol plays an important role in the kidney of the one-humped Arabian camel during chronic dehydration, we searched for signs of TFs involved in the regulation of the synthesis, storage, degradation and/or transport of cholesterol and other lipids. Interestingly, amongst the DE TFs shared by camels and jerboas, we found the Sterol Regulatory Element Binding TFs 1 (*SREBF1*) and 2 (*SREBF2*). Their corresponding proteins, SREBP1 and

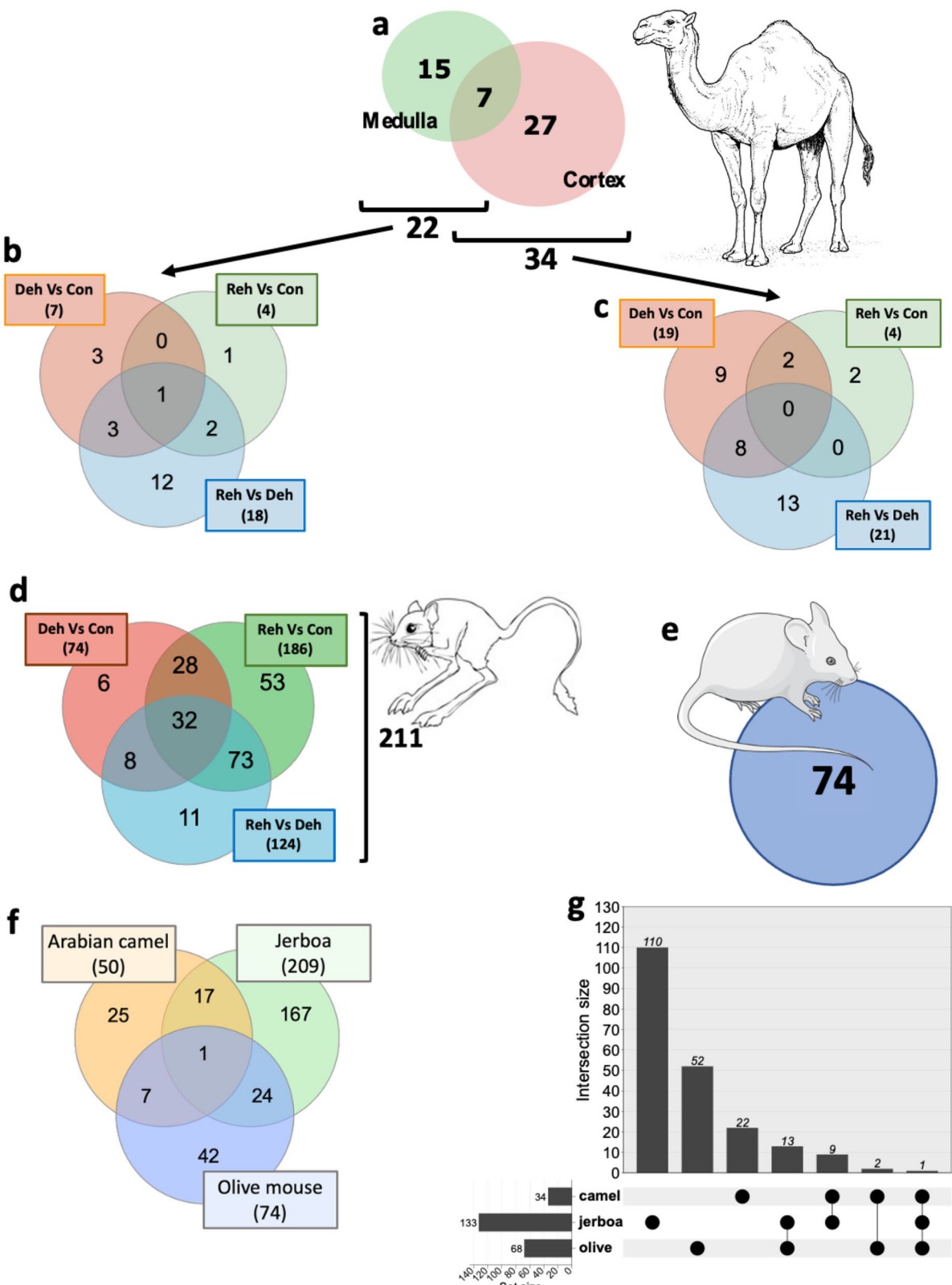

**Fig. 1 | Differentially expressed transcription factors in the kidney of 3 species adapted to water deprivation and the overlap between species. a** We identified 49 differentially expressed transcription factors in the kidney of the one-humped Arabian camel during chronic dehydration and subsequent acute rehydration, 27 in the cortex, 15 in the medulla and 7 co-regulated in both tissues. The number of DE TFs according to condition in the cortex and the medulla separately are shown in (**b**) and (**c**), respectively. **d** We further identified 251 DE TFs in the kidney of jerboas during chronic dehydration and subsequent acute rehydration. **e** Differentially expressed transcription factors identified in the kidney of olive mice living in a xeric environment compared to individuals inhabiting wet areas. **f** Venn plot showing the overlap between DE TFs identified in camels, olive mice and jerboas. **g** Upset plot showing the overlap of all differentially enriched regulons characterised using RTN between the three species. Representations of the animals were obtained from a Public Domain Archive.

SREBP2, are basic helix-loop-helix leucine zipper TFs that regulate the biosynthetic pathway of fatty acid and cholesterol by stimulating transcription of genes containing sterol-response-elements[10]. We knew that genes downstream of these TFs were downregulated from our previous work, so we decided to investigate the regulatory network of the SREBP proteins. Moreover, even though it is not a direct TF, *INSIG1*, encoding an endoplasmic reticulum membrane protein that regulates cholesterol metabolism, lipogenesis, and glucose homeostasis, and an intrinsic part of this regulatory complex, was significantly downregulated in our RNAseq data. Thus, we further investigated the regulation of the cholesterol biosynthesis process via the INSIG1-SREBP1/SREBP2 complex using RT-qPCR for validation of gene expression at the transcript level and immunohistochemistry for protein visualization in camels and jerboas. We also analyzed the differential expression of SREBP1 and SREBP2 target genes.

In the cortex of the dromedary kidney, RT-qPCR validation showed that *INSIG1*, *SREBF1* and *SREBF2* were significantly downregulated in dehydration compared to controls. The expression values for these genes returned to control levels following rehydration (Fig. 2a). Publicly available ChIP-Seq data attributes 966 targets to SREBP2, with 63 being DE in the cortex of the dromedary. According to condition, 39 targets were DE in dehydration compared to controls, 2 were DE in rehydration compared to controls and 42 were DE in rehydration compared dehydration (Fig. 2b, c). Similarly, 706 target genes have been linked to SREBP1 and 53 were DE in the kidney cortex. Twenty-seven targets were DE in dehydration compared to controls, 11 were DE in rehydration compared to controls and 31 were DE in rehydration compared dehydration (Fig. 2e, f). GO analyses of the DE target genes revealed an enrichment of several categories related to cholesterol/lipids synthesis and metabolism (Fig. 2d, g). We experimentally validated *IDI1*, a target of SREBP2, which coding enzyme is involved in the production of a substrate of the cholesterol biosynthesis pathway. *IDI1* was significantly downregulated ($p_{adj} < 0.05$) in dehydrated camels. Together with *INSIG1* (SREBP1 target) and *SQLE*[2] (target of SREBP2), which were already validated, we have validated target genes of SREBP1 and SREBP2 that are actively involved in the cholesterol biosynthesis pathway and significantly change in our RNAseq data (Supplementary Fig. SF6). In the kidney medulla, only *SREBF2* was significantly downregulated during dehydration compared to controls. Changes in differential expression of *INSIG1* and *SREBF1* did not reach the significance threshold but showed a similar trend. No differences were found between rehydrated and control camels (Fig. 2h). In this tissue, we identified 77 DE target genes for SREBP2; 19 in dehydration compared to controls, 9 in rehydration compared to controls and 64 in rehydration compared to dehydration (Fig. 2i, j). Likewise, 48 target genes of SREBP1 were DE in our RNAseq data; 18 in dehydration compared to controls, 12 in rehydration compared to controls and 31 in rehydration compared to dehydration (Fig. 2l, m). GO analyses of the target genes also showed enrichment of cholesterol/lipids synthesis and metabolism in the kidney medulla (Fig. 2k, n). In this case, *IDI1* was also significantly downregulated, while ACSS2 (target of SREBP1 and SREBP2) was significantly upregulated in rehydration ($p_{adj} < 0.05$). This gene is also required for the synthesis of cholesterol (Supplementary Fig. SF6).

In the kidney of the jerboa, our RNAseq data showed a significant regulation of *SREBF1* and *SREBF2* in dehydrated and rehydrated animals compared to controls, however, RT-qPCR showed non-significant changes across conditions, although *SREBF2* does seem to be downregulated (Fig. 3a). Nonetheless, we analysed the expression of SREBP1 and SREBP2 target genes in the kidney of the jerboa. A total of 408 SREBP2 targets were DE in the kidney of jerboas; 125 in dehydration compared to controls, 361 in rehydration compared to controls and 252 in rehydration compared to dehydration (Fig. 3b, c). Similarly, 294 SREBP1 targets were DE in the kidney of jerboas; 100 in dehydration compared to controls, 255 in rehydration compared to controls and 192 in rehydration compared to dehydration (Fig. 3e, f). GO analyses showed a significant enrichment of terms related to cell metabolism but also to cellular response to stress, cell death and apoptosis (Fig. 3d, g). In jerboa kidney, we investigated *APOE* and *FOXO3* expression by RT-qPCR. APOE is a key mediator of cholesterol

transport and plays an important role in cholesterol homeostasis, while FOXO3 functions as a trigger of apoptosis. Unfortunately, differences between conditions did not reach the significance threshold, likely do to having subtle changes in the RNAseq data and the already discussed large variability. Nevertheless, APOE seems to increase after rehydration, while FOXO3 decreases during dehydration, tendencies that are in accordance with our line of thought. Additionally, we compared fold change after dehydration and rehydration between qPCR results and RNAseq data for all genes that shows good correlation, showing that the tendencies we describe are consistent (Supplementary Fig. SF6).

We did not identify changes in the expression levels of either *INSIG1*, *SREBF1* or *SREBF2* in the kidney of the olive mouse between animals from the xeric Patagonian steppe compared to individuals from the wetlands of the Magellanic and Valdivian forests. Neither sign of regulation of the cholesterol biosynthesis pathway in the GO analyses, and thus, we did not further pursue this analysis.

### Immuno-localisation of SREBP1, SREBP2 and INSIG1 during chronic dehydration and acute rehydration in camels and jerboas

Using immunofluorescence techniques, we investigated the localisation of SREBP1, SREBP2 and INSIG1 in the kidney of camels and jerboas. SREBP1 and SREBP2 are synthesised in the endoplasmic reticulum (ER) and then transported via vesicles to the Golgi apparatus where they are cleaved before the mature protein is trafficked to the nucleus. Therefore, we used RCAS1 as Golgi marker to pinpoint the localisation of SREBP1 and SREBP2 during chronic dehydration and acute rehydration.

In camels, the localisation and the signal intensity of SREBP1 remained relatively constant during dehydration and rehydration compared to controls in both cortex and medulla (Fig. 4).

SREBP2 was found to co-localise with RCAS1 in the cortex in control animals, but it seems more packed during dehydration and rehydration. Interestingly, in the medulla, SREBP2 was present in most of the nuclei in controls but localised mostly perinuclear during dehydration. The nuclear signal appeared to increase with rehydration (Fig. 5). Thus, despite a significant downregulation of *SREBF1* and *SREBF2*, only the cellular localisation of SREBP2 seems to be affected. Perhaps SREBP2 is retained in the Golgi apparatus during dehydration to reduce the activation of the cholesterol synthesis pathway. The localisation of INSIG1 did not change in the kidney of the one-humped Arabian camel during dehydration nor during rehydration (Supplementary Fig. SF7).

In jerboas (Fig. 6), the localisation and signal intensity of SREBP1 seem to be similar in all treatment groups, similar to that in camel. Renal SREBP2 signal, however, seems to be reduced in dehydrated animals compared to control and rehydrated jerboas. SREBP2 and RCAS1 seem to co-localise in dehydrated animals; however, this may not be so clear in the control and the rehydrated groups. Again, as in the case of camels, the protein might be retained in the Golgi apparatus during water deprivation to limit the activation of the cholesterol synthesis pathway, although in this case such localisation in and out of the nuclei was not clear. Signal intensity appeared similar to control after rehydration. The localisation of INSIG1 did not change either in the kidney of the jerboa during dehydration nor during rehydration (Supplementary Fig. SF8).

### Discussion

Next-generation RNA sequencing (RNAseq) has become the preferred method to study gene expression[11,12], providing accurate quantification of mRNAs. However, data analysis remains challenging given the large amount of data that is generated and its complexity. Thus, many studies are prone to focus on specific genes of their interest, or to rely on enrichment analyses, such as Gene Ontology (GO)[13], or a combination of both approaches, to try to make biological sense of the data. These are excellent tools to unravel biological functions, pathways and/or processes that may be of interest in a given condition, but complete data mining is, nonetheless, largely out of reach. Further, computational tools to process RNAseq data were developed for organisms with well-studied reference genomes and

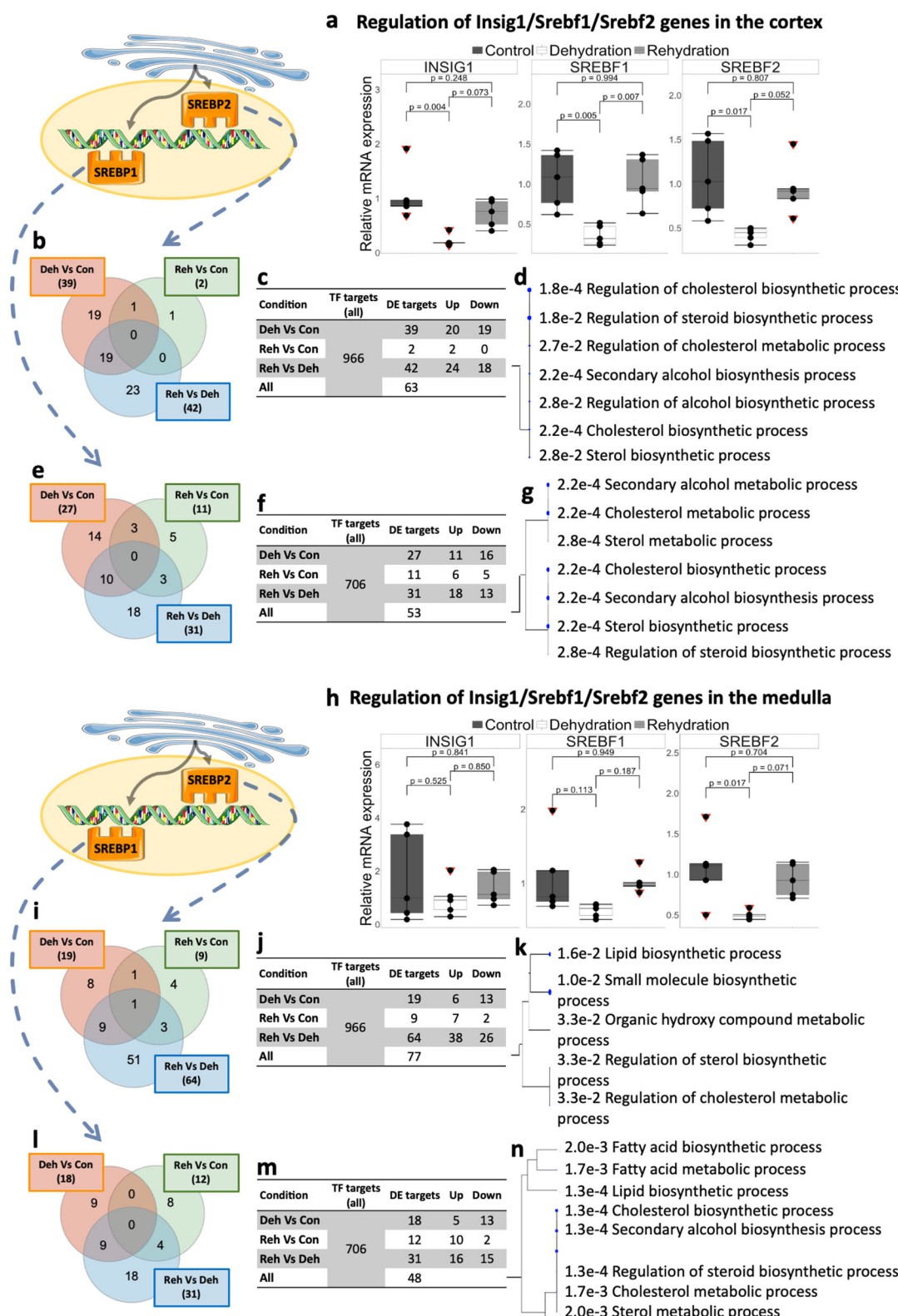

**a  Regulation of Insig1/Srebf1/Srebf2 genes in the cortex**

**h  Regulation of Insig1/Srebf1/Srebf2 genes in the medulla**

transcriptomes. However, working with incomplete references, which is often the case in studies of non-model organisms, may introduce errors and omissions when determining expression levels[14].

In this work, we used the RTN package to reconstruct the regulatory network of 3 desert-adapted species under different hydration regimes and further mined previous RNAseq data. This approach allowed us to identify

TFs and their targets. This may give additional information about the changes that take place in the kidney under specific hydration conditions. The Reconstruction of the TRNs and the Analysis of Regulons (RTN)[10,15] is used to infer regulons and to estimate the level of enrichment of a regulatory network. More importantly, this tool does not rely on significance thresholds but use log fold changes of every transcript detected in an experiment to

**Fig. 2 | Transcriptional regulatory activity of the INSIG1-SREBP1/SREBP2 complex in the kidney of the one-humped Arabian camel.** Relative changes in gene expression of *INSIG1*, *SREBF1* and *SREBF2* in the Arabian camel kidney **a** cortex and **h** medulla after chronic dehydration and acute rehydration compared to controls. SREBPs are escorted to the Golgi where they are cleaved and translocated back to the nucleus where they bind to sterol response elements (SREs). Comparison of the means by one-way ANOVA (Tukey's post hoc correction). The boxplots are presented with the S.E.M ($n = 5$), centre lines show median, box edges delineate 25 and 75th percentiles and bars extend to minimum and maximum values. Individual data points represent biologically independent samples and data points within red triangles denote outliers, all the outliers highlighted were included for the statistical analyses (RT-qPCR source data in Supplementary Data SD3). Differentially expressed target genes of SREBP2 (**b, c, i, j**) and SREBP1 (**e, f, l, m**) in different conditions and number of gene targets that were upregulated and downregulated according to RNAseq data over all previously ChIP-Seq identified targets are also shown. Gene Ontology analyses of differentially expressed target genes in the cortex and the medulla are shown for SREBP2 (**d, k**) and SREBP1 (**g, n**). Images were created from previously-created icons, all icons were freely available to download, modify and use in scientific publications from SMART Servier Medical Art under the terms of the Creative Commons Attribution 4.0 International License (https://smart.servier.com/) and from REACTOME under the Creative Commons Attribution 3.0 Unported License (https://reactome.org/icon-lib).

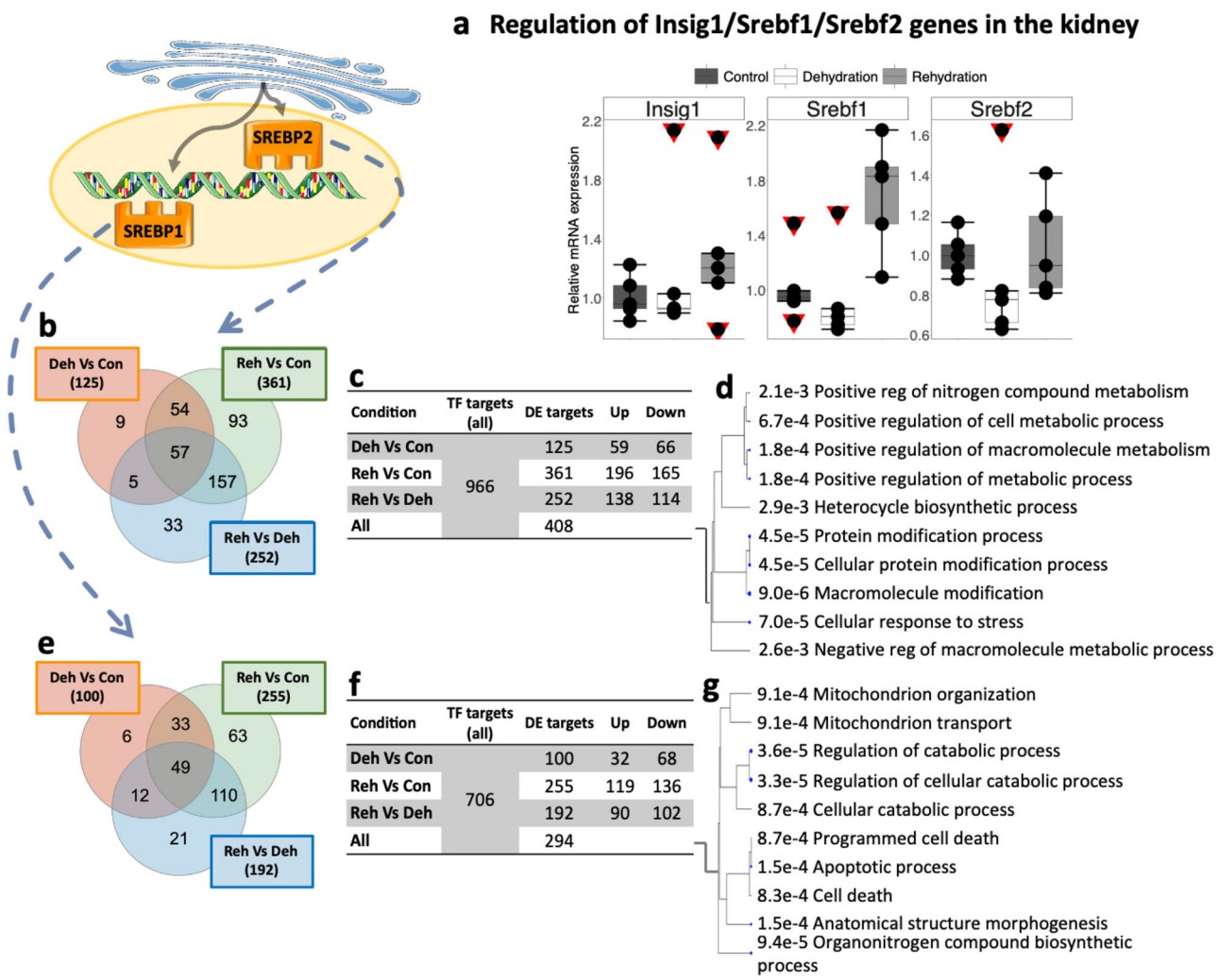

**Fig. 3 | Transcriptional regulatory activity of the INSIG1-SREBP1/SREBP2 complex in the kidney of the jerboa. a** Relative changes in gene expression of *INSIG1*, *SREBF1* and *SREBF2* in the jerboa kidney after chronic dehydration and acute rehydration compared to controls. SREBPs are escorted to the Golgi where they are cleaved and translocated back to the nucleus where they bind to sterol response elements (SREs). Comparison of the means by one-way ANOVA (Tukey's post hoc correction). The boxplots are presented with the S.E.M ($n = 5$), centre lines show median, box edges delineate 25th and 75th percentiles and bars extend to minimum and maximum values. Individual data points represent biologically independent samples and data points within red triangles denote outliers, all the outliers highlighted were included for the statistical analyses (RT-qPCR source data in Supplementary Data SD3). Differentially expressed target genes of SREBP2 (**b, c**) and SREBP1 (**e, f**) in different conditions and number of gene targets that were upregulated and downregulated according to RNAseq data over all previously ChIP-Seq identified targets are also shown. Gene Ontology analyses of differentially expressed target genes are shown for SREBP2 (**d**) and SREBP1 (**g**). Images were created from previously-created icons, all icons were freely available to download, modify and use in scientific publications from SMART Servier Medical Art under the terms of the Creative Commons Attribution 4.0 International License (https://smart.servier.com/) and from REACTOME under the Creative Commons Attribution 3.0 Unported License (https://reactome.org/icon-lib).

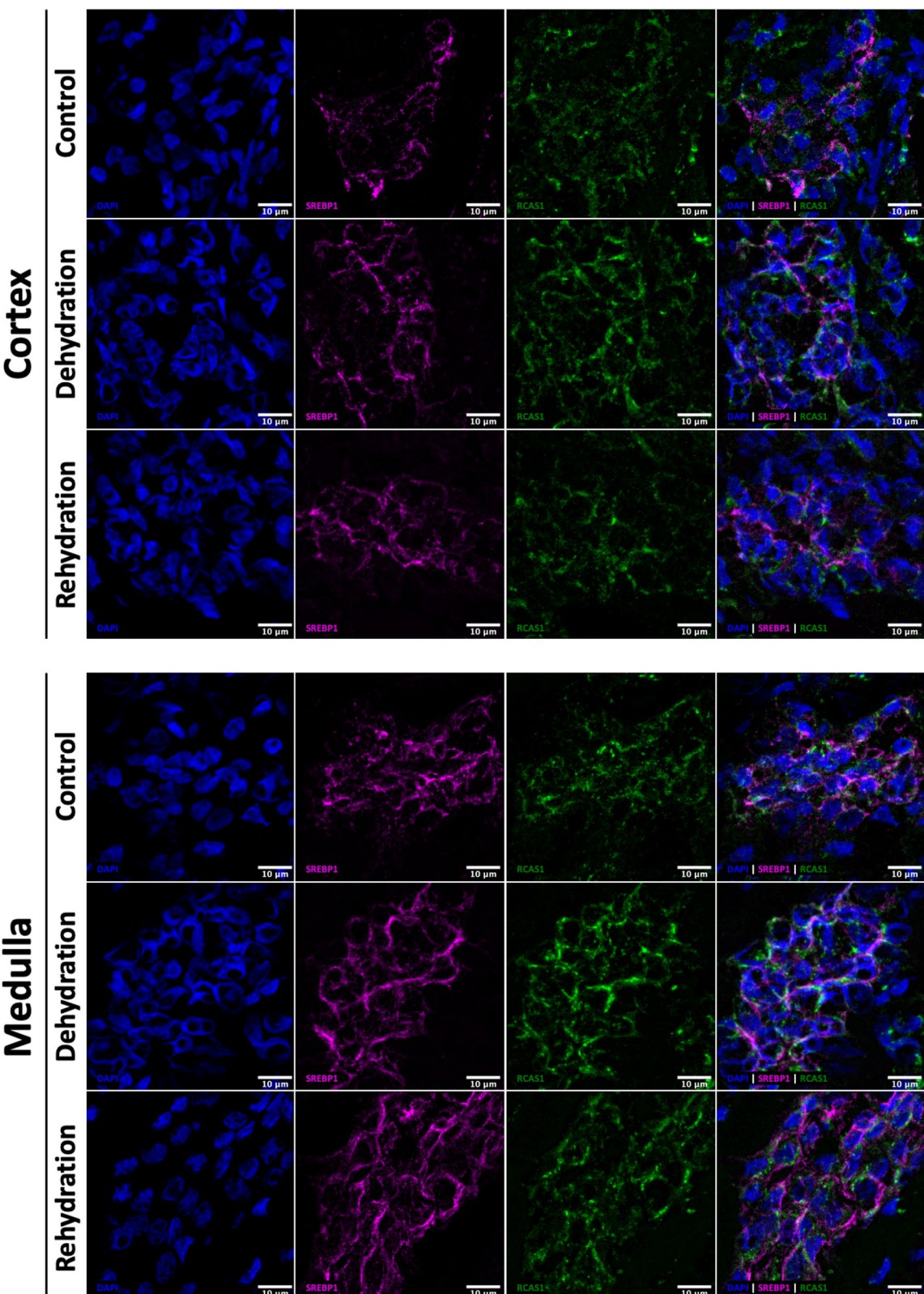

**Fig. 4 | Cellular localisation of SREBP1 in the kidney of the one-humped Arabian camel during chronic dehydration and acute rehydration.** Immunofluorescence staining of Sterol Regulatory Element-Binding Protein 1 (SREBP1) in camel kidney cortex (top panel) and medulla (bottom panel) sections from controls, dehydrated and rehydrated animals. Images are representative of several cross sections of each kidney compartment. Localisation of SREBP1 (magenta) is shown relative to the nuclei (DAPI, blue, 1st column) and the Golgi apparatus (RCAS1, green, 3rd column) in the overlap images (4th column). Scale bars, 10 μm.

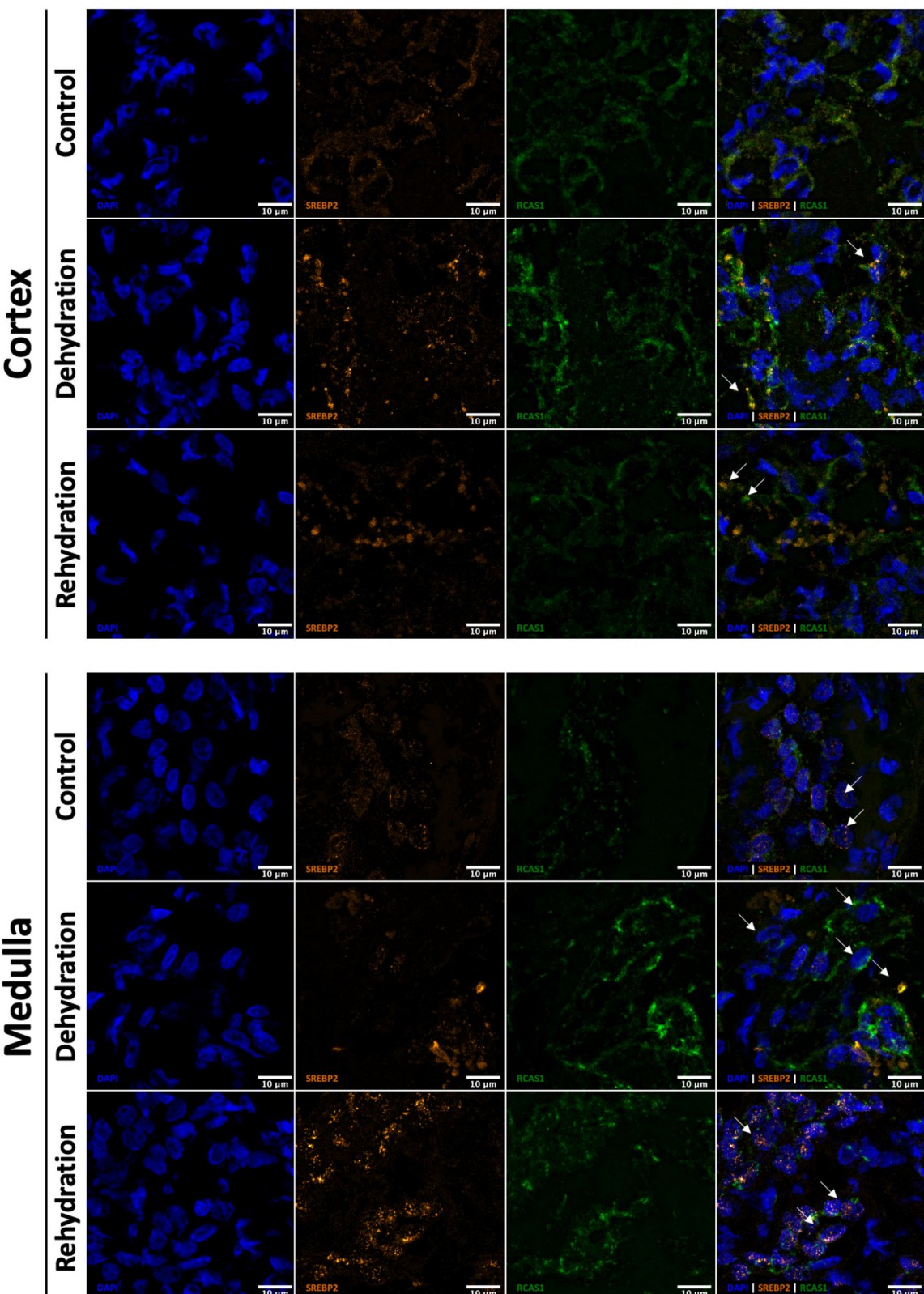

**Fig. 5 | Cellular localisation of SREBP2 in the kidney of the one-humped Arabian camel during chronic dehydration and acute rehydration.** Immunofluorescence staining of Sterol Regulatory Element-Binding Protein 2 (SREBP2) in camel kidney cortex and medulla sections from controls, dehydrated and rehydrated animals. Images are representative of several cross sections of each kidney compartment.

Localisation of SREBP2 (orange) is shown relative to the nuclei (DAPI, blue, 1st column) and the Golgi apparatus (RCAS1, green, 3rd column) in the overlap images (4th column). Arrows on the overlay images indicate the localisation of SREBP2 relative to the cellular markers DAPI and RCAS1 under the 3 experimental conditions. Scale bars, 10 μm.

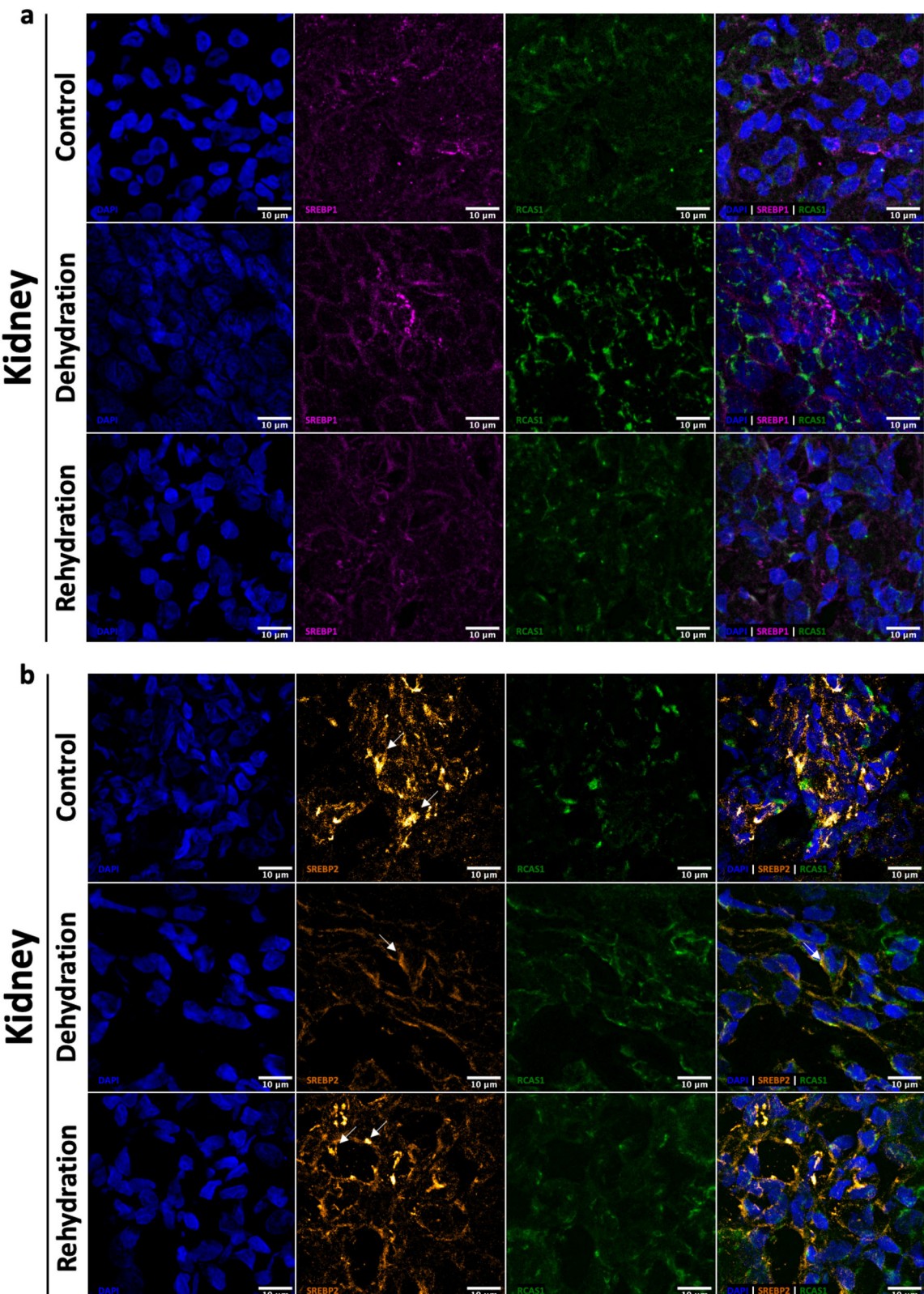

**Fig. 6 | Cellular localisation of SREBP1 and SREBP2 in the kidney of the jerboas during chronic dehydration and acute rehydration.** Immunofluorescence staining of Sterol Regulatory Element-Binding Proteins 1 (SREBP1) (**a**) and 2 (SREBP2) (**b**) in jerboa whole kidney sections from controls, dehydrated and rehydrated animals. Images are representative of several cross sections of the kidney. Localisation of SREBP1 (magenta) and SREBP2 (orange) is shown relative to the nuclei (DAPI, blue, 1st column) and the Golgi apparatus (RCAS1, green, 3rd column) in the overlap images (4th column). Arrows indicate the localisation of SREBP2. Scale bars, 10 μm.

identify enriched networks potentially important in the system that would otherwise be missed even when these genes do not always reach the significance threshold when considered individually. This technique is particularly useful to investigate non-model species when the genome annotation is incomplete, there is lack of appropriate reference assemblies[14] and large variability is often unavoidable[16], all of which may introduce errors in the calculations of expression levels. In these situations, statistical estimations may be affected, and the presence of different gene isoforms and alternative splicing events adds a level of complexity that makes comparisons with model species challenging and potentially biased. Thus, analyses of the TRNs based on RNAseq data are useful to further explore the data and to reveal key genes that were a priori not relevant or unchanged. Further, together with GO analyses, this method allows comparing different species in a simpler way because TFs tend to be very well conserved, and their networks might have similar functions, which are readily identified using gene set functional analyses such as GO. In spite of this, most TFs and their regulons are understudied[7], and most investigations have focused attention on human disease and development processes. This is even more so in non-model organisms where, to the best of our knowledge, there are no studies that have investigated TRNs associated with specific physiological conditions.

We have catalogued all DE TFs and, interestingly, uncovered significantly enriched regulons which TFs and/or targets did not pass the significance threshold in the RNAseq analysis, but may be important to cope with water deprivation, in all 3 species. For instance, we found that *NFE2L1*, which targets regulating cholesterol removal, is not DE according to our RNAseq data, but its overall regulon is significantly enriched in both camels and jerboas. A similar example is *TFCP2L1*, which is involved in the development of the salt-water homeostasis-regulating principal cells of the collecting ducts. Many other pathways and regulatory networks seem to be enriched in all three species; future research intended to explore this line of research may use these initial observations as the base of their work. However, we decided to focus on further investigating the cholesterol biosynthesis pathway to fully characterise our previous findings[2]. In short, we demonstrated in that work a role for cholesterol during chronic dehydration. By analysing the transcriptomes and the proteomes of the camel kidney during dehydration and after rehydration, we found that several enzymes involved in the cholesterol biosynthesis pathway were downregulated. We validated those findings and revealed a decrease in membrane cholesterol in the kidney of dehydrated camels. We argued that ion and water transport may be indirectly enhanced by the suppression of the cholesterol biosynthetic process, and the subsequent reduction in membrane cholesterol. Interestingly, we found that three key TFs involved in the cholesterol biosynthesis process (SREBP1, SREBP2 and INSIG1) were co-regulated in camel and jerboa. Further, our recent work showed histopathological and ultrastructural changes in camel kidney after long-term dehydration using light and electron microscopy[17]. Unfortunately, we could not run this analysis in jerboas. Briefly, dehydrated camels showed fatty degeneration and focal necrosis in glomeruli, as well as lesions in proximal and distal convoluted tubules. Electron microscopy revealed, among others, damaged endothelial fenestrations, abnormal chromatin display, abnormal cuboidal cells, ruptured nuclear membranes, vacuolation and degenerated organelles. In general, lesions improved with rehydration. Effects were somewhat smaller in the medulla. While these animals showed significant changes in many dehydration markers, nephropathy was not observed, so we decided to further explore the cholesterol biosynthesis pathway using immunohistochemical techniques to localise SREBP1, SREBP2 and INSIG1.

While INSIG1 and SREBP1 were possibly not affected during dehydration, SREBP2 seemed to be retained in the Golgi apparatus during dehydration. We argue that this may be a mechanism to limit the traffic of this protein into the nuclei, thus capping the activation of the cholesterol synthesis pathway (Fig. 7). The role for this pathway is supported by the latest research, which describe the cholesterol biosynthesis pathway as a convergent evolution hotspot including a key genomic variant in *INSIG1* that seems reasonably well-conserved in desert species among 22 species

of ungulates[18], and a conserved deletion in *HMGCR* affecting the sterol-sensing domain[19] that plays an important role in cholesterol biosynthesis. Obviously, reducing the amount of cholesterol in the cell can only go so far since it is an essential compound for multiple processes and structures. Thus, a partial knockdown of the pathways fits well with our findings. We further identified several targets of these TFs that were DE in our RNAseq data and directly involved in the regulation of cholesterol in camels and cell damage and apoptosis in jerboas. These pathways have been previously identified in studies using kidney cells by other authors[20–23]. These data further supported our previous findings and reinforced our hypothesis for a role of cholesterol during dehydration in camels[1,2]. Although we are confident with our hypothesis, we acknowledge that this work lacks a direct link between water deprivation and the regulation of SREBP1 and SREBP2. Further research is needed to fully characterise the molecular basis of this mechanism.

Despite some canonical mechanisms involved in water conservation have developed in most species, others must have necessarily diverged. The RNAseq data of olive mouse[24] shows about 3000 DE genes in animals living in the Patagonian steppe compared to those living in rain forest and many of them were involved in "classical" water conservation pathways in addition to others newly identified. Thus, we argue that the reduction of cholesterol may simply not be in the repertoire of this species.

Nevertheless, limitations of this methodology must be considered. First of all, for practical reasons during animal selection, only male individuals were included in these investigations. Adding female animals would certainly improve the conclusions of the study. Then, TRN models describe and predict relationships between molecular structures, for instance, a set of genes that interact with each other to control a specific cell function. There are several computational ways to infer TRNs. Information Theory models (e.g., ARACNE) are the most commonly used methods due to their computational simplicity and a relatively good performance with a low number of samples. Moreover, using transcriptomic data to reconstruct TRNs is generally preferred over other datasets due to its ability to establish gene-gene interactions[9]. However, due to the complexity of running studies with camels, our sample size is small and that may affect the performance of the RTN analysis. Regarding the immunofluorescence images, the study will benefit from segment-specific images to finely pinpoint the localisation of SREBP1, SREBP2 and INSIG1, however, this was out of the scope of the study, and the samples available only allowed running the experiments in whole kidney slices. Furthermore, specific ChIP-Seq experiments for our TF candidates would strengthen our conclusions. Finally, large variability detected during qPCR validations also impacts the interpretation of the data, as we thoroughly discussed in our previous work[2]. Thus, further investigations may be needed to further characterise the TRNs involved in water conservation in camels and jerboas.

In this study, we have comprehensively described the TRNs of the one-humped Arabian camel kidney cortex and medulla and the jerboa kidney during severe dehydration and subsequent acute rehydration. We revealed several TFs and regulatory networks potentially involved in coping with water deprivation and identified candidates for validation. We went on to study TFs involved in the cholesterol biosynthesis pathway, which were directly linked to our previous findings[1,2]. Our datasets suggested that SREBP2, the TF preferentially involved in the activation of the cholesterol synthesis pathway, and many of its target genes, are differentially regulated during dehydration in camels and jerboas. We further observed changes in the localisation and signal intensity of SREBP2 using immunofluorescence techniques. Considering all data, and in accordance with our previous work, we argue that SREBP2 is actively regulated and directly involved during the response of camels and jerboas to acute dehydration.

## Methods
### Animals
Nineteen male dromedary camels aged 4–5 years, body weight range 276–416 kg, were used in the present study[2]. As previously described[2], the camels were supplied with alfalfa hay as feed and ranch-housed in the

**Fig. 7 | The regulatory machinery of the cholesterol synthesis pathway.** In normal hydration conditions, INSIG1, SREBP1 and SREBP2 regulate the activation of the cholesterol synthesis pathway. In camels (top panel), their transcripts are down-regulated during dehydration, although effects at protein level for INSIG1 and SREBP1 were not found. However, SREBP2 was potentially being retained in the Golgi apparatus in order to limit the cholesterol biosynthesis pathway. In jerboas (bottom panel), only *SREBF2* was downregulated. Again, more SREBP2 seems to be retained in the Golgi apparatus. Images were created from previously-created icons, all icons were freely available to download, modify and use in scientific publications from SMART Servier Medical Art under the terms of the Creative Commons Attribution 4.0 International License (https://smart.servier.com/) and from REACTOME under the Creative Commons Attribution 3.0 Unported License (https://reactome.org/icon-lib). Representations of the animals were obtained from a Public Domain Archive.

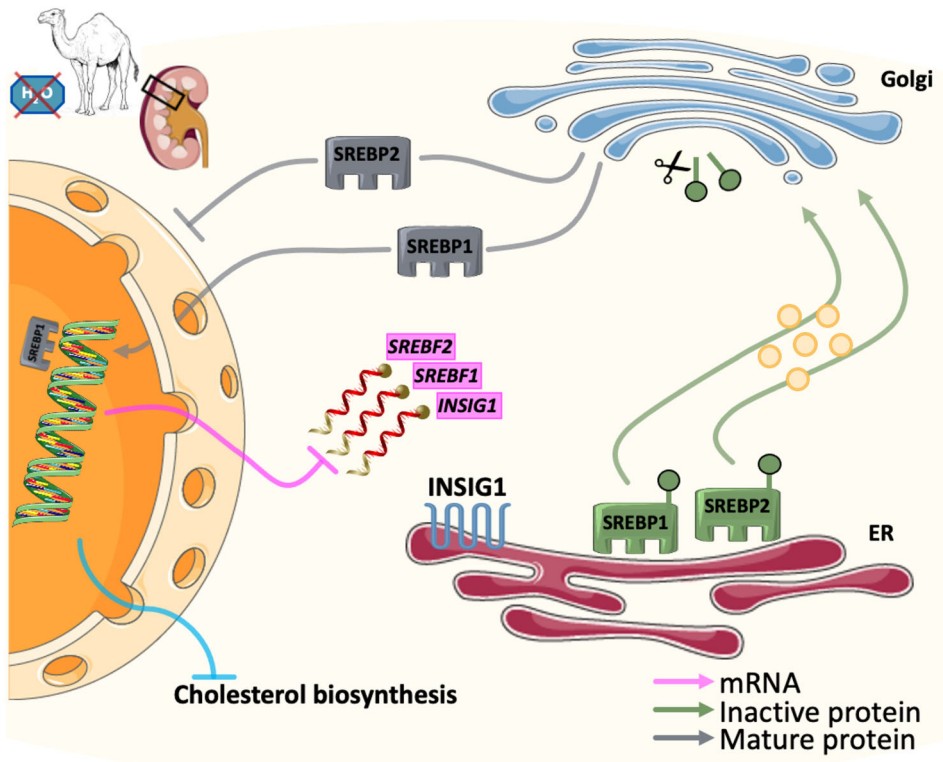

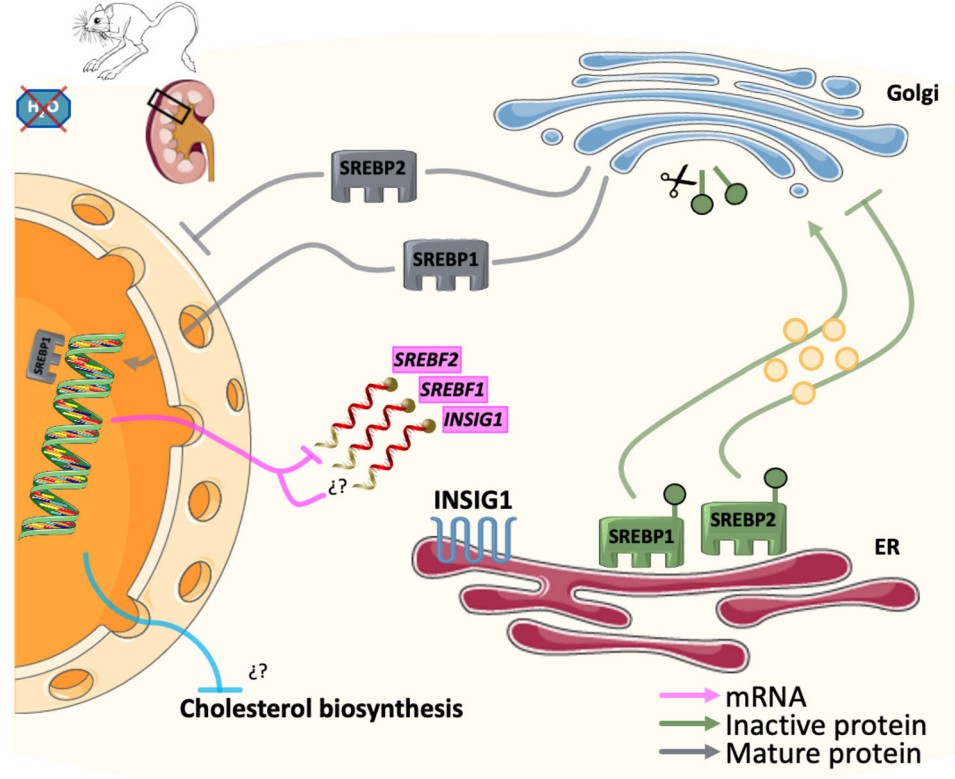

United Arab Emirates. Veterinary supervision was provided throughout the experimental period and no signs of distress or illness were identified. After a short adaptive period, the camels were divided into three groups, control ($n = 5$), dehydrated ($n = 8$) and rehydrated ($n = 6$). The control group had free access to food and water during the entire experimental period. The dehydrated group was water-deprived but had *ad libitum* access to food for 20 days. Meanwhile the rehydrated group was subjected to the same protocol as the dehydrated animals followed by *ad libitum* water supply for 3 additional days. After the experimental period, the camels were sacrificed in the local central abattoir for human consumption in April 2016. Kidney

samples were harvested within 1 hour after killing the animals, immediately frozen in liquid nitrogen and kept at $-80\,°C$ for later physiological, histological, morphological, and molecular analysis[2]. Samples were shipped frozen on dry ice to the University of Bristol under the auspices of a DEFRA Import Licence (TARP/2016/063).

Wild male jerboas ($n = 15$) were caught in the area surrounding Riyadh, Saudi Arabia[1]. Age of the animals cannot be reported as they were wild, but they were considered adults. As described in our previous work[1], all animals were collected in January-March 2019 and were kept single housed, 25 °C constant temperature, 12 h light/12 h dark photoperiod for acclimatisation for a minimum of 14 days. They were then randomly divided into 3 experimental groups: euhydrated ($n = 5$), dehydrated ($n = 5$), and rehydrated ($n = 5$). All groups were fed, weighed, and had their cages cleaned daily. The euhydrated group had access to water ad libitum for the duration of the study, the dehydrated group had their water source removed for 10 days and the rehydrated group had their water removed for 7 days and then had access to water ad libitum for 5 additional days. Animals were sacrificed by ether overdose whilst still in their housing to avoid unnecessary stress from handling. Kidneys were dissected and frozen with dry ice[1]. All extractions were performed immediately after sacrifice and between 08:30 am and 12:30 pm. Samples were shipped frozen on dry ice to the University of Bristol under the auspices of a DEFRA Import Licence (ITIMP18.0082).

A total of 38 wild, adult Patagonian olive mice were captured using rodent Trap Special-S traps (Forma Ltda., Santiago, Chile) at 4 different locations, 2 localities in the dry Patagonian steppe and another 2 within the Valdivian and Magellanic humid forests area as described in Giorello et al.[3,24]. Exact age is not reported as they were wild animals. Briefly, animals were weighed and euthanized following the guidelines of the American Society of Mammalogists, by properly trained personnel. Then, animals were ventrally dissected, and the right kidney of every individual was carefully excised, weighed and stored in liquid nitrogen. All methods involving Abrothrix olivacea were carried out as part of the review process for the Fondecyt Research Grant 1110737[24].

It is worth noting that dehydration times were different. Camels' weight is orders of magnitude higher than those of rodents and, therefore, dehydration time must be adjusted to comply with ethical regulations. Despite this fact, canonical markers for dehydration were measured in the original works, and we are confident to say that the hydration status at each time point is well defined and that transcriptomic analyses are robust. In any case, we have not run statistical analysis to make inter-species comparisons because multiple confounding variable would apply. We only looked at the overlapping DE genes to potentially identify commonalities or conserved pathways between desert species.

## Transcriptomic analysis

Total RNA extractions were conducted from kidney tissue, using Zymo Direct-Zol RNA miniprep (R2052, Zymo) as per manufacturer's instructions, of 15 randomly selected camels[2] and 15 jerboas[1], as previously reported. RNeasy mini kit (74004, Qiagen) was used to extract RNA from the kidneys of 38 olive mice following recommendations of the manufacturer[3,24]. In all experiments, RNA quantity and purity were assessed with NanoDrop, and Poly-A selection mRNA enrichment and paired-end library preparation were performed for Illumina platform bulk RNA sequencing.

RNAseq analysis pipeline for camels and jerboas was described in Alvira et al.[2] and Gillard et al.[1]. Briefly, as we reported in those works, an in-house computer was used to process sequencing data from camels and jerboas using a bespoke pipeline. First, reads were trimmed of adaptor sequences using BBDuk tool followed by FastQC[25] quality control. STAR[26] was used to map reads using default settings to the publicly available *Camelus dromedarius* (GCA_000803125.2; CamDro2) and *Jaculus jaculus* (GCA_000280705.1; JacJac 1.0) genome assemblies downloaded from the Ensembl 100 database[27]. Mapped reads were summarised using FeatureCounts[28] grouping to gene identifiers. In olive mouse, a single node-machine was used to process the data. RNAseq reads were mapped against

the reference assembly[3] using Bowtie2 aligner. Mapped reads were also summarised as feature counts for each gene identifier. DESeq2[29] in Rv4.0.1[30] was used to estimate differential expression of genes between groups in all 3 experiments[3,24].

## Reconstruction of the TRN (RTN)

Firstly, we used biomaRt[31,32] function in Rv4.0.1[30] to draw all the GO categories associated to each of the DE genes identified after RNAseq analyses. Then, using GO:0003700 (DNA-binding transcription factor activity), we filtered and selected genes previously described as TFs.

Further, we used the RTN package[15,33] in R to identify significantly enriched TRNs. The package tests the association between a given TF and all potential targets using RNAseq data. It is tuned to deal with large gene expression datasets to build transcriptional regulatory units that are centred on TFs. Briefly, the Transcriptional Network Interference (TNI) takes in a matrix of normalised gene expression data, the corresponding gene and sample annotation and the list of TFs to be analysed to construct a TNI-class object. The TNI permutation function processes the TNI-class object by permutation analysis and returns inferred TRNs. The TNI permutation function was run using 2000 permutations. Unstable interactions are subsequently removed by bootstrap analysis and weak TF-TF interactions were filtered using the data processing inequality theorem (dpi). The Transcriptional Network Analysis (TNA) pipeline was then used to do enrichment analysis over the previously identified regulons[15]. Firstly, the TNI-class object was converted into a TNA-class object, which consists of a named numeric vector with log2 fold changes from the DESeq2 analysis ranked from highest to lowest, a character vector listing the DE genes and a data frame with gene annotations mapped to the vector containing log2 fold change values. Then, we used a two-tailed gene set enrichment analysis (GSEA-2T)[33,34] to test whether the regulons are positively or negatively associated to the expression data. GSEA-T2 was run using 2000 permutations. The goal is to assess whether the target genes are overrepresented among the genes that are more positively or negatively DE. A running sum increases every time a gene from the ranked list is part of the regulon and decreases when it is not. The enrichment score is stablished by the maximum distance of the running score from the x-axis. A large positive enrichment score indicates an activated regulon, while a large negative enrichment score indicates a repressed regulon.

## Gene validation

We validated genes of interest by RT-qPCR in one-humped Arabian camel and jerboa kidneys. Total RNA was extracted as described above. cDNA was synthesised (GoScript$^{TM}$ Reverse Transcriptase, Promega) from 1 μg of total RNA following the manufacturer's protocol. Then, following the same protocol reported earlier [2], RT-qPCRs were run using StepOne real-time qPCR system (Thermo Fisher Scientific) using PowerUp SYBR Green Master Mix (Thermo Fisher Scientific) in 10 μl reactions (final volume). We designed transcript-specific primer pairs using Primer-BLAST (NCBI)[35] and reference genomes for dromedary and jerboa. One-humped Arabian camel genome assembly[36] is available at NCBI under the accession number PRJNA269274 and RNAseq data is accessible through GEO Series accession number GSE173683 at https://www.ncbi.nlm.nih.gov/geo/query/acc.cgi?acc=GSE173683 [2]. Jerboa genome assembly[37] may be found in Ensembl (https://www.ensembl.org/Jaculus_jaculus/Info/Index) under the accession number GCA_000280705.1 and RNAseq data is accessible through GEO Series accession number GSE225470 at https://www.ncbi.nlm.nih.gov/geo/query.acc.cgi?acc=GSE225470[1]. We generated standard curves for each primer pair from serial dilutions (2-fold) and calculated $R^2$ and reaction efficiencies using the equation $E = 10\text{-}(1/\text{slope})\text{-}1$[38]. We finally determined transcription levels of these genes by calculating deltadeltaCt change corrected for reaction efficiency. Primer sequences are shown in Supplementary Table 1. *PPIA* was selected as housekeeping gene after we shortlisted potential candidates[39]. RT-qPCR source data is available in Supplementary Data SD3.

## Immunohistochemistry

We investigated the expression of SREBP1, SREBP2 and INSIG1 at protein level using immunohistochemical staining. Briefly, 16 μm thick kidney sections from camel (cross-sections of cortex and medulla separately) and jerboa (cross-section of cortex and medulla) were incubated in ice-cold 4% paraformaldehyde (PFA; VWR, USA, cat. number 28794.295) in phosphate buffered saline (PBS; Sigma, USA, cat. number P3813-10PAK, Lot SLCH0992) solution for 15 min in a slide dipper for fixation. Then, slides were washed in PBS 3×5 minutes followed by a 20 min blocking in 3% Bovine Serum Albumin (BSA; Sigma-Aldrich, USA, cat. number A7906-50G, Lot SLCH3826) in 0.3% Triton X-100 (Sigma, USA, cat. number T8787-100ML, Lot MKBF335) prepared in PBS (PBS-t). Primary antibodies anti-mouse SREBP-1 (diluted 1:50 in 1% BSA PBS-t; Santa Cruz Biotechnology, Inc., sc-365513, Lot L0420), anti-mouse SREBP-2 (diluted 1:50 in 1% BSA PBS-t; Santa Cruz Biotechnology, Inc., sc-13552, Lot J0820) and anti-mouse INSIG-1 (diluted 1:50 in 1% BSA PBS-t; Santa Cruz Biotechnology, Inc., sc-390504, Lot L2116), anti-rabbit RCAS1 as Golgi marker (diluted 1:150 in 1% BSA PBS-t; Cell Signalling Technology, Inc., D6P5J, Lot:1) were added and the slides left incubating overnight, at 4 °C, on a levelled rack with the bottom covered with PBS to avoid drying of the samples. After the incubation, slides were washed 3 × 5 min in PBS and then incubated with the secondary antibodies Alexa Fluor™ 488 donkey anti-mouse (diluted 1:500 in PBS-t; Life Technologies, USA, cat. number A21202, Lot 2018296) and Alexa Fluor™ 633 donkey anti-rabbit (diluted 1:500 in PBS-t; SIGMA, USA, SAB4600132, Lot 18C0920) for 1 h at room temperature followed by 3 × 5 min PBS washes. A final 15 min incubation with DAPI (stock solution 1 μg/ml in PBS, diluted 1:1000) was followed by a final wash in PBS before mounting the slides using Fluoroshield (Sigma, USA, cat. number F6182-20ML, Lot MKCL6078). Images were acquired using a confocal microscope (Leica SP5-II confocal laser scanning microscope attached to a Leica DMI 6000 inverted epifluorescence microscope) at the Wolfson Bioimaging Centre and analysed using ImageJ2/Fiji Software[40].

## Statistics and reproducibility

As in Alvira et al.[2], Benjamini-Hochberg was used to calculate significance levels in differential gene expression analysis. DE genes (DEGs) are defined as those showing significant differential expression between 2 conditions, where significance threshold is reached when the adjusted p value is below 0.05. Statistical significance between RT-qPCR experimental groups was calculated using One-way ANOVA with Tukey's post hoc test for multiple pairwise comparisons. Alternatively, we used Kruskal-Wallis test combined with Benjamini-Hochberg method for groups, which followed non-normal distribution. All the outliers highlighted in the boxplots were included for the statistical analyses. These statistical analyses were performed intra-species. It is important to note that this was not an attempt to pair-wise compare the 3 species, since they are fairly different in many ways (body mass, body size, body shape, behaviour, life style…) and it would not be possible in this case to account for all confounding variables, but a way to depict TFs and pathways potentially conserved in species adapted to water deprivation. Samples sizes (n) are described in the main text for every experiment/analysis. RT-qPCR measurements for every sample were run in duplicates. All statistical tests were run using R software. $p_{adj} < 0.05$ was considered significant.

## Ethical

The study with camels was approved by the Animal Ethics Committee of the United Arab Emirates University (approval ID: AE/15/38) and the University of Bristol Animal Welfare and Ethical Review Board. The study with jerboas was approved by the Animal Ethics Committee of the Khalifa University and the University of Bristol Animal Welfare and Ethical Review Board. Finally, all methods involving *Abrothrix olivacea* were carried out in accordance with a protocol reviewed and approved by the Ethics Committee of the Fondo Nacional de Ciencia y Tecnología (FONDECYT, Chile) and the Ethics Committee of the Universidad Austral de Chile (UACh, Chile). Thus, we have complied with all relevant ethical regulations for animal use.

## Reporting summary

Further information on research design is available in the Nature Portfolio Reporting Summary linked to this article.

## Data availability

The transcriptomic data underlying these analyses, including raw FASTQ files, bulk RNAseq counts, DESeq2 data and project metadata, were deposited in NCBI's Gene Expression Omnibus (GEO)[41] or NCBI Sequence Read Archive (SRA)[42] as part of previous publications as follows. One-humped Arabian camel genome assembly[36] is available at NCBI under the accession number PRJNA269274 and RNAseq data is accessible through GEO Series accession number GSE173683 at https://www.ncbi.nlm.nih.gov/geo/query/acc.cgi?acc=GSE173683 [2]. Jerboa genome assembly[37] may be found in Ensembl (https://www.ensembl.org/Jaculus_jaculus/Info/Index) under the accession number GCA_000280705.1 and RNAseq data is accessible through GEO Series accession number GSE225470 at https://www.ncbi.nlm.nih.gov/geo/query/acc.cgi?acc=GSE225470 [1]. Olive mouse data was also deposited in NCBI, and reads are available under BioProject accession number PRJNA471316. Reference transcriptome assembly is available through Dryad at https://doi.org/10.5061/dryad.7nh50k7. SREBP1 and SREBP2 target genes lists were downloaded from ChIP-X database[43] which is freely available at http://amp.pharm.mssm.edu/lib/chea.jsp. The source data behind the figures in the paper are available in Supplementary Data SD3. All other data are available from the corresponding authors upon reasonable request.

## Code availability

All software and code used to analyse the RNAseq data is described in the Methods section, has been previously described in the literature and are common, well-established bioinformatic tools used in omics studies, as previously reported[1,2]. The custom Python script used to assembly the olive mouse reference transcriptome is available in Giorello et al.[24]. The scripts used to run the RTN analyses are described in Fletcher et al.[15] and Castro et al.[33].

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

## Acknowledgements

This research was generously supported by grants from the Leverhulme Trust (RPG-2017-287) to B. T. Gillard, F. Alvira-Iraizoz, D. Murphy and M. P. Greenwood, and the United Arab Emirates University (UAEU)-Programme for Advanced Research (UPAR-31M242) to A. Adem. The Austrian Science Funds (P29623-B25) supported P. A. Burger. A. Alagaili was funded by the Researchers Supporting Project number (RSPD2025R602), King Saud University. Students were supported by grants from the Biotechnology and Biological Sciences Research Council-SWBio DTP programme (BBSRC BB/M009122/1) to B. T. Gillard, the Medical Research Council (MRC 1662603) to A. Paterson, and the British Heart Foundation (BHF FS/17/60/33474) to A. G. Pauža. The authors gratefully acknowledge the Wolfson Bioimaging Facility for their support and assistance in this work.

## Author contributions

D. M., M. P. G. and F. A.-I. conceived the study. D. M., M. P. G., A. A. and A. N. A. equally supervised the project. M. P. G. and B. T. G. collected the camel samples. D. M., M. P. G., B. T. G. and F. A.-I. designed the experiments. B. T. G. and F. A.-I. analysed the transcriptomic data and performed bioinformatic analyses. F. A.-I. and A. G. P. adapted the script and run the RTN analyses of all species. F. A.-I. and P. L. run all immunohistochemical analyses. F. A.-I. wrote the manuscript. A. P. performed read alignment and data curation. P. B. performed the reference genome assembly and assisted with bioinformatic advice. N. A. performed animal work and collected the jerboa samples. M.H.A. performed laboratory work and data analysis with camel samples. All authors contributed to revise the manuscript.

## Competing interests

The authors declare no competing interests.
