## [Transparent Peer Review file · Communications Biology]

Reconstruction of the transcriptional regulatory networks in the kidney of desert-adapted species

Corresponding Author: Professor David Murphy

Version 0:

Reviewer comments:

Reviewer #1

(Remarks to the Author)

Firstly, the paper presents a reconstruction of the transcriptional regulatory networks (TRNs) of the kidneys of the Arabian camel, jerboa, and olive mouse under different water regimes, specifically during dehydration and subsequent rehydration. This reconstruction provides a comprehensive view of the gene expression changes and regulatory interactions that occur in these animals' kidneys in response to water stress.

Secondly, the study identifies a significant number of differentially expressed transcription factors (TFs) and regulons in each species. These findings contribute to our understanding of the molecular mechanisms underlying the animals' physiological adaptations to water stress.

Thirdly, the paper confirms a role for cholesterol during dehydration in camels and jerboas, and investigates the localization of key molecules involved in cholesterol synthesis, such as SREBP1, SREBP2, and INSIG1, in the kidneys of these animals. The results suggest that the regulatory machinery of the cholesterol synthesis pathway is affected in the kidneys of these animals during dehydration.

Regarding the novelty and interest of these claims, the reconstruction of TRNs under different water regimes is a relatively novel approach in the field of transcriptome analysis, particularly in the context of animal adaptations to extreme environments. The identification of differentially expressed TFs and regulons, as well as the investigation of cholesterol metabolism in these animals, provides new insights into the molecular mechanisms of physiological adaptation to water stress. These findings are likely to be of interest to researchers in the fields of comparative physiology, genomics, and evolutionary biology, as they contribute to our understanding of how animals cope with environmental challenges.

However, I have some questions here: as follows

Point 1: Please rewrite the abstract and introduction section, including background, purpose, brief description of materials, methods, results, and significance.

Point 2: Why choose species camel, jerboa, and olive mouse instead of other species, and what are their commonalities? How did the author consider analyzing them together?

Point 3: Suggest adding a section to the discussion on how to reduce the interference caused by species differences during data analysis.

Point 4: During the dehydration process of animals, their metabolic pathways undergo a series of adaptive changes in response to water loss and physiological stress? The author only focuses on the cholesterol synthesis pathway here, what about other pathways?

Point 5: In the method, the dehydration time of camels and rodents is not consistent, and the degree of kidney dehydration cannot be explained consistently, which may cause errors in subsequent sequencing analysis.

Point 6: Please include dehydrated kidney tissue slices in the manuscript to support your data analysis.

Point 7: Based on the presentation results in the author's manuscript, it cannot directly indicate that the signal pathway in the figure 7 is related to renal dehydration.

Reviewer #2

(Remarks to the Author)

The authors have conducted a meaningful exploration by investigating RNA alterations in the kidneys of Arabian camels and jerboas during water deprivation and interpreting partial results from the perspective of transcription factors. This is a valuable study with visually appealing figures. However, before recommending publication, several issues require more detailed elaboration:

Rationale for selecting SREBF1 and SREBF2: The justification for choosing SREBF1 and SREBF2 as focal points in this study needs further clarification.

Pathological observations in Figures 5 and 6: The authors should provide standard pathological staining (e.g, HE) to illustrate the renal morphological changes induced by dehydration.

Link between cholesterol metabolism and dehydration: The association between cholesterol metabolism and water deprivation requires more comprehensive description and discussion.

Validation of transcription factor activity: As transcription factors, the downstream targets regulated by SREBF1 and SREBF2 should be experimentally validated. For instance, qPCR could be used to confirm whether changes in downstream gene expression align with the sequencing results.

Version 1:

Reviewer comments:

Reviewer #1

(Remarks to the Author)

The manuscript is well-structured, with logical flow between sections. Figures and tables are clearly labeled and effectively support the text. The discussion contextualizes findings within the broader literature while acknowledging limitations, which strengthens the credibility of the work.

Reviewer #2

(Remarks to the Author)

The author has basically answered all the relevant questions and it is recommended to publish this article.

COMMSBIOL Revisions – Rebuttal letter for Referees

Manuscript number: COMMSBIO-25-0366

We would like to thank the Reviewers for their robust and scholarly assessment of our manuscript. The suggested revisions have greatly improved the paper. All changes are highlighted in yellow in the revised manuscript and are addressed below point-by-point.

Review #1

Point 1: Please rewrite the abstract and introduction section, including background, purpose, and brief description of materials, methods, results, and significance.

We agree that the abstract and the introduction were missing context regarding dehydration, physiological adaptation, etc... We have now addressed this.

Firstly, we adapted the abstract. Please note that given the limited space available (150 words), we could not add as much detail as we would have liked. We nevertheless have included context in the abstract as suggested. Line 24

Regarding the introduction, we have added a short paragraph to give context without extending the length of the introduction too much.

Line 37: Climate change associated desertification and extreme drought are posing an unprecedented burden on desert-dwelling species. However, some species have evolved to thrive under these conditions. A plethora of adaptations to life in the desert have been described in desert-dwelling species over the past decades. More recently, molecular and genomic approaches have started to unravel the underlying mechanisms of these adaptations. So far, multiple mechanisms and metabolic pathways have been associated to better adaptation to water scarcity and extreme environmental conditions.

Line 76: State of the purpose of the study.

We aimed to explore the TRNs involved in chronic dehydration in desert-adapted species.

Point 2: Why choose species camel, jerboa, and olive mouse instead of other species, and what are their commonalities? How did the author consider analysing them together?

The reason to choose these species was two fold. On the one hand, the collaboration between the authors was first established to study camel brain and kidney transcriptomics during dehydration and subsequent acute rehydration. This initial work was, from our view, very successful. The results of these investigations were published in *Communications Biology* in 2021 and 2022. Then, we had the possibility to carry on similar work using jerboa as model species (data published in 2023 in *iScience*). The idea behind this study was to investigate species of animals very well-adapted to withstand water deprivation for prolonged periods of time. Both species, being obviously very different, fit this criteria. On the other hand, transcriptomic data of olive mouse was freely available from a previous work by Giorello et al. We contacted the author and he was keen for us to use his data. This looked particularly interesting because they studied 2 subpopulations of olive mouse, one living in a xeric environment and another living in a rain forest. In this case, differences were established between subpopulations rather than between treatments. This is briefly explained in the manuscript, but we hope these additional lines help clarifying the context.

Commonalities are clear, improved capacities to retain water and thrive in desert conditions. Importantly, although we present overlapping transcription factors / regulatory networks, we did not attempt to analyse them together. In other words, we did not compare them pair-wise. This would not make sense because the animals and environments are very different (size, body mass, body shape, behaviour, environmental conditions...), which we believe is the concern of the reviewer. However, we considered it interesting to show the overlapping features to depict conserved pathways and commonalities.

Highlight in line 154.

Point 3: Suggest adding a section to the discussion on how to reduce the interference caused by species differences during data analysis.

As mentioned in the response to point 2, we did not compare them pair-wise. This would not make sense because the animals and environments are very different (size, body mass, body shape, behaviour, environmental conditions...), which we believe is the concern of the reviewer. However, we considered it interesting to show the overlapping features to depict conserved pathways and commonalities.

We have added a few words in the statistics section to clarify this. Line 620.

Point 4: During the dehydration process of animals, their metabolic pathways undergo a series of adaptive changes in response to water loss and physiological stress? The author only focuses on the cholesterol synthesis pathway here, what about other pathways?

This is a fair point; we would have liked to discuss all pathways, but the analysis revealed way too many to discuss them all. Just to mention a couple: many genes involved in water conservation are differentially regulated in all species; in jerboa, pathways related to apoptosis and cell death are enriched; in camels, cholesterol as mentioned and oxidative stress in the mitochondria also undergo modifications.

We believe that including all these data will make the manuscript too confusing and difficult to read and it will lose focus.

That said, we published very interesting data regarding the cholesterol biosynthesis pathway and considered appropriate following that line of research. In addition, all data was deposited in a repository and is available for anyone keen to run their own analyses and further investigate interesting pathways.

That said, we briefly discuss other interesting findings in paragraph starting in line 378 and have added a sentence for clarity.

Point 5: In the method, the dehydration time of camels and rodents is not consistent, and the degree of kidney dehydration cannot be explained consistently, which may cause errors in subsequent sequencing analysis.

We acknowledge that, perhaps, we have not been able to explain this sufficiently.

Necessarily, dehydration time for camels and rodents must be different. Whilst a camel weighs around 300kg, a jerboa weighs about 60 grams. It is already quite impressive that jerboas undergo 10-11 day dehydration without severe consequences, comparable to camels that are deprived of water for 21 days despite being a lot heavier and capable of keeping a much lower relative metabolism. Given the body weight, it would be unethical to dehydrate rodents for longer. Just as an example, ethical committees would not allow keeping rats under a water deprivation regime for more than 2-3 days.

We have added a line in the methods section to clearly state this issue. Line 513. More importantly, we reiterate that no statistical comparison was done between species, since we are well aware of the multiple confounding variables that would apply. We also state this in our response to Point 2 and Point 3. We have only looked at genes that are differentially expressed in all species (overlap) to potentially identify genes or pathways conserved in desert species or that are common to these species, but no statistical comparison was done. Thus, sequencing analysis is not compromised, we run it individually for each species.

The degree of dehydration may not be the same, but several canonical markers for dehydration were measured to characterise the hydration status.

Point 6: Please include dehydrated kidney tissue slices in the manuscript to support your data analysis.

We appreciate this comment since we failed to cite our findings regarding kidney histopathology (light and electron microscopy). This work was published in BMC Veterinary Research in 2024 (Damir et al. Effects of long-term dehydration and quick rehydration on the camel kidney: pathological changes and modulation of the

expression of solute carrier proteins and aquaporins. BMC Vet Res 20, 367 (2024). <https://doi.org/10.1186/s12917-024-04215-4>).

We have cited this article in the discussion and briefly described the main results. Line 391. We considered adding this information in the results to give context to our finding and to figures 5 and 6, as recommended by Reviewer #2, but since it is not data generated specifically for this paper, we believed it fits better in the discussion.

Point 7: Based on the presentation results in the author's manuscript, it cannot directly indicate that the signal pathway in the figure 7 is related to renal dehydration.

We have added a line to state that a direct link is missing, although we are confident with the results.

Line 417. "Although we are confident with our hypothesis, we acknowledge that this work lacks a direct link between water deprivation and the regulation of SREBP1 and SREBP2. Further research is needed to fully characterise the molecular basis of this mechanism."

Review #2

Point 1: Rationale for selecting SREBF1 and SREBF2: The justification for choosing SREBF1 and SREBF2 as focal points in this study needs further clarification.

We have added a line in the results section to link between the search of transcripts involved somehow in the cholesterol biosynthesis pathway and the decision for choosing SREBP1 and SREBP2.

Line 178. Their corresponding proteins, SREBP1 and SREBP2, are basic helix-loop-helix leucine zipper transcription factors that regulate biosynthetic pathway of fatty acid and cholesterol by stimulating transcription of genes containing sterol-response-elements. We knew that genes downstream of these TFs were downregulated from our previous work, so we decided to investigate the regulation of the SREBP proteins.

Point 2: Pathological observations in Figures 5 and 6: The authors should provide standard pathological staining (e.g, HE) to illustrate the renal morphological changes induced by dehydration.

Reviewer #1 raised the same concern:

We appreciate this comment since we failed to cite our findings regarding kidney histopathology (light and electron microscopy). This work was published in BMC Veterinary Research in 2024 (Damir et al. Effects of long-term dehydration and quick rehydration on the camel kidney: pathological changes and modulation of the expression of solute carrier proteins and aquaporins. BMC Vet Res 20, 367 (2024). <https://doi.org/10.1186/s12917-024-04215-4>).

We have cited this article in the discussion and briefly described the main results. Line 391. We considered adding this information in the results to give context to our finding and to figures 5 and 6, as recommended by Reviewer #2, but since it is not data generated specifically for this paper, we believed it fits better in the discussion.

Point 3: Link between cholesterol metabolism and dehydration: The association between cholesterol metabolism and water deprivation requires more comprehensive description and discussion.

The link between cholesterol and dehydration in camels is extensively discussed in our first paper of the topic (Alvira-Iraizoz, F., Gillard, B.T., Lin, P. et al. Multiomic analysis of the Arabian camel (*Camelus dromedarius*) kidney reveals a role for cholesterol in water conservation. Commun Biol 4, 779 (2021). <https://doi.org/10.1038/s42003-021-02327-3>).

For clarity, we have briefly described in the manuscript our main finding and reasoning regarding the link between the cholesterol biosynthesis pathway and dehydration, as suggested by the Reviewer. Line 382.

Point 4: Validation of transcription factor activity: As transcription factors, the downstream targets regulated by SREBF1 and SREBF2 should be experimentally validated. For instance, qPCR could be used to confirm whether changes in downstream gene expression align with the sequencing results.

First of all, we would like to say that we completely agree with the Reviewer. We thus made an effort to identify the most interesting targets, at least in the context of our line of research, and validate them using qPCR. Thus, we validated different SREBP1 and SREBP2 targets in camel and jerboa to support our methodology. Lines 202, 217 and 247. And Supplementary Figure SF6.

Additional comments:

We have added the following information related to recent publications that we have considered important to include:

Line 405: The role for this pathway is supported by the latest research, which describes the cholesterol biosynthesis pathway as a convergent evolution hotspot including a key genomic variant in *INSIG1* that seems reasonably well-conserved in desert species among 22 species of ungulates¹⁸, and a conserved deletion in *HMGCR* affecting the SREBP cleavage domain¹⁹ that plays an important role in cholesterol biosynthesis.

COMMSBIOL Revisions – Rebuttal letter for Referees

Manuscript number: COMMSBIO-25-0366B

Dear Referees / Colleagues:

We would like to thank you, once again, for your robust and scholarly assessment of our manuscript. As we have mentioned also to the Editors, your comments and suggestions helped to improve the article, and all authors are very pleased with the final version.

Since you did not rise further comments and/or suggestions, the manuscripts stays fundamentally as it is. We only want to report a few modifications suggested by the editor, namely:

1. Extend a little bit the abstract to elaborate on the background and the main findings.
2. Add references for genome and transcriptome reference assemblies.
3. Add oligos sequences.
4. Include an ethical statement.

Thank you.

Sincerely,

Fernando Alvira-Iraizoz (on behalf of all authors)